# Reverse-Annealed Sequential Monte Carlo for Efficient Bayesian Optimal Experiment Design

**Jake Callahan**[*]
Program in Applied Mathematics
The University of Arizona

**Andrew Chin**[*]
Department of Biostatistics
Johns Hopkins University

**Jason Pacheco**
Department of Computer Science
The University of Arizona

**Tommie Catanach**
Computational Data Science
Sandia National Laboratories

## Abstract

Expected information gain (EIG) is a crucial quantity in Bayesian optimal experimental design (BOED), quantifying how useful an experiment is by the amount we expect the posterior to differ from the prior. However, evaluating the EIG can be computationally expensive since it generally requires estimating the posterior normalizing constant. In this work, we leverage two idiosyncrasies of BOED to improve efficiency of EIG estimation via sequential Monte Carlo (SMC). First, in BOED we simulate the data and thus know the true underlying parameters. Second, we ultimately care about the EIG, not the individual normalizing constants. Often we observe that the Monte Carlo variance of standard SMC estimators for the normalizing constant of a single dataset are significantly lower than the variance of the normalizing constants across datasets; the latter thus contributes the majority of the variance for EIG estimates. This suggests the potential to slightly increase variance while drastically decreasing computation time by reducing the SMC population size, which leads us to an EIG-specific SMC estimator that starts with only a single sample from the posterior and tempers *backwards* towards the prior. Using this single-sample estimator, which we call reverse-annealed SMC (RA-SMC), we show that it is possible to estimate EIG with orders of magnitude fewer likelihood evaluations in three models: a four-dimensional spring-mass, a six-dimensional Johnson-Cook model and a four-dimensional source-finding problem.

## 1 Introduction

Optimal experimental design (OED) is a powerful method for selecting design parameters for experiments that update model uncertainty using observational data. By quantifying the utility $U$ of a design $d$, one can maximize the utility over all the designs as: $d^* = \arg\max_d U(d)$. In Bayesian optimal experimental design (BOED) we are interested in the information gain (IG) from an experiment for an unknown parameter $\theta$ given the dataset $y$ (Lindley, 1956). This is quantified by the Kullback-Leibler (KL) divergence from the prior to the posterior (Rainforth et al., 2024):

$$\mathrm{IG}(y \mid d) = D_{KL}(p(\theta \mid y, d) \parallel p(\theta)) = \int_\theta p(\theta \mid y, d) \log \frac{p(\theta \mid y, d)}{p(\theta)} \, \mathrm{d}\theta. \tag{1}$$

---

[*]Denotes equal contribution. These authors are listed in alphabetical order.

39th Conference on Neural Information Processing Systems (NeurIPS 2025).

As we do not have access to $y$ before an experiment is run, our utility function is the expected information gain (EIG):

$$\text{EIG}(d) = \mathbb{E}_{y|d}[D_{KL}(p(\theta \mid y, d) \parallel p(\theta))] = \int_y \int_\theta p(\theta, y \mid d) \log \frac{p(y \mid \theta, d)}{p(y \mid d)} \, d\theta \, dy \qquad (2)$$

This expectation has no analytical solution outside of the most basic examples, and thus one typically resorts to Monte Carlo integration. This proceeds by drawing a dataset, estimating the IG, and repeating this many times to obtain an EIG. However, the IG itself is also generally intractable due to the presence of the posterior normalizing constant $p(y \mid d)$, also known as the model evidence or marginal likelihood. Thus, considerable effort is often placed on estimating $p(y \mid d)$ or its logarithm (Ryan, 2003), and not all methods are guaranteed to give accurate results. In low dimensions, the simple nested Monte Carlo (NMC) is a common choice (Rainforth et al., 2018; Zheng et al., 2018).

Higher dimensions or more informative data require more stable, but expensive, Monte Carlo estimators. Principal among these is a class of estimators based on sequential Monte Carlo (SMC) (Del Moral et al., 2006; Chopin et al., 2020). SMC starts with many samples from a known distribution and evolves those samples through a sequence of intermediate distributions toward the distribution of interest. Likelihood values computed during this evolutionary process can then be used to compute the model evidence (Xie et al., 2011). Often, hundreds of millions of costly likelihood evaluations are required to compute the EIG for a single design, making SMC infeasible for many models.

Our work adapts and extends the bidirectional Monte Carlo approach introduced by Grosse et al. (2015) for the specific context of BOED by leveraging two unique aspects of the problem. First, we observe that in BOED we require $\mathbb{E}[\log p(y)]$, and the variance in $\log p(y)$ across different $y$ often dominates the variance in estimating $\log p(y)$ for a single $y$ when using SMC. We find that we can significantly reduce the number of particles to achieve similar Monte Carlo error but with far less computational cost. The second observation is that generating $y$ requires drawing $\theta \sim p(\theta)$ and then $y \sim p(y \mid \theta)$, resulting in the joint sample $(\theta, y) \sim p(\theta, y)$. Instead of discarding $\theta$, we can treat this joint sample as if we drew $y \sim p(y)$ and $\theta \sim p(\theta \mid y)$, giving us a free posterior sample. Together, these two facts allow us to modify the framework proposed in Grosse et al. (2015), which starts with a single posterior sample and draws from a sequence of distributions tempered backwards towards the prior, and then estimates an upper bound on $\log p(y)$ accordingly.

This approach yields *reverse-annealed sequential Monte Carlo* (RA-SMC), an algorithm that starts with a single posterior sample and draws from a sequence of distributions tempered backward towards the prior. Unlike Grosse et al. (2015), which sandwiches the true value of $\log p(y)$, RA-SMC directly provides practical EIG estimates and remains robust to overestimating under poor MCMC mixing. We demonstrate this estimator on a coupled spring-mass system, Johnson-Cook model of plastic deformation, and a sequential source location problem–all with multimodal posteriors. Not only do we show that traditional SMC estimators can be used with an order of magnitude fewer particles, but also that our reverse estimator provides a further fourfold improvement in computational cost.

## 2 Related Work

EIG estimation in BOED has been approached with various methods. *Nested Monte Carlo* (NMC) uses inner-outer loops (Ryan, 2003; Beck et al., 2018) but scales pooorly and has slow-decaying bias (Rainforth et al., 2018; Zheng et al., 2018). *Variational* approaches optimize an approximate posterior (Dahlke et al., 2023; Foster et al., 2019, 2020); they scale better but can be costly and miss complex posterior structure (Pacheco & Fisher, 2019). *Likelihood-free* methods bypass liekelihood via density-ratio estimation, as in LFIRE (Kleinegesse et al., 2021), or by directly learning the objective, as in MINEBED (Kleinegesse & Gutmann, 2020). For sequential design, *reinforcement learning* can learn a design policy (Huan & Marzouk, 2016; Foster et al., 2021; Ivanova et al., 2021; Blau et al., 2022).

Asymptotically consistent alternatives include tempering methods like *annealed importance sampling* (AIS) (Neal, 1996). *Sequential Monte Carlo* (SMC) extends this by resampling and rejuvenating particles along the temperature sequence (Chopin et al., 2020; Del Moral et al., 2006), which has been used for evidence estimation (Xie et al., 2011) and BOED (Ryan et al., 2016). We note here that in this work, "sequential" refers to the progression of particles through a sequence of tempered distributions, as distinct from the temporal sequence in state-space and filtering applications. Our

estimator is a single-particle, zero-resample instance of SMC tempering. Unlike AIS, our particle is equally weighted at each temperature and rejuvenated via MCMC, making it conceptually closer to MCMC.

*Bidirectional Monte Carlo* (BDMC) (Grosse et al., 2015, 2016) runs forward and reverse annealing to produce stochastic upper and lower bounds on $\log p(y)$a. A reverse pass starts from an exact posterior sample and a forward pass from the prior, "sandwiching" the marginal likelihood for model comparison. We adopt BDMC's use of a free posterior draw and reverse anneal but omit the forward pass and bounding machinery.

Our estimator can be regarded as a modification of the BDMC framework for the BOED context. While BDMC "sandwiches" the marginal likelihood, we repurpose the reverse-annealing pass the create a practical EIG estimator. Crucially, we use the thermodynamic integral instead of BDMC's stepping stone algorithm to mitigate the bias of reverse estimators. Furthermore, while BDMC requires many particles for tight bounds, our analysis of EIG's variance structure shows a single particle provides sufficient accuracy at a fraction of the compuational cost. To our knowledge, this is a novel application of reverse-annealed SMC to BOED and constitutes a new EIG estimator.

The methods described above present a diverse set of trade-offs between computational cost, scalability, and asymptotic guarantees. This work specifically focuses on improving the efficiency of tempering-based Monte Carlo estimators, like AIS and SMC, which provide asymptotically-exact estimates crucial for high-fidelity applications, and we show that the RA-SMC estimator makes this powerful but expensive class of methods more practical. To that end, we first review the standard SMC framework, which forms the basis of our proposed approach.

## 3 Preliminaries: Annealed Sequential Monte Carlo

We briefly review the SMC framework (Chopin et al., 2020) and the SMC-based stepping stone algorithm (Xie et al., 2011) that is commonly used for computing model evidence $p(y)$. SMC evolves a population of $N$ particles from a simple initial distribution (e.g., the prior) to the posterior. It does so by progressing through a sequence of intermediate tempered distributions using importance resampling and Markov chain Monte Carlo (MCMC) steps. One commonly-chosen sequence of tempered distributions is the power posterior sequence (Friel & Pettitt, 2008):

$$p_{t_i}(\theta \mid y) = \frac{p(y \mid \theta)^{t_i} p(\theta)}{z_{t_i}(y)}, \quad \text{where,} \quad z_{t_i}(y) = \int_\theta p(y \mid \theta)^{t_i} p(\theta) \, d\theta \qquad (3)$$

and $0 = t_0 \leq \cdots \leq t_N = 1$ is an increasing sequence of temperatures. The lowest temperature $t_0$ recovers the prior, and at the highest temperature $t_N$, the posterior. At each level, we can compute:

$$E[p(y \mid \theta)^{\Delta t_i}] = \int_\theta p(y \mid \theta)^{\Delta t_i} \frac{p(y \mid \theta)^{t_i} p(\theta)}{z_{t_i}} \, d\theta = \frac{\int_\theta p(y \mid \theta)^{t_{i+1}} p(\theta) \, d\theta}{z_{t_i}} = \frac{z_{t_{i+1}}(y)}{z_{t_i}(y)} \qquad (4)$$

where $\Delta t_i = t_{i+1} - t_i$. Taking the product up to the $N - 1$ level yields the marginal likelihood $p(y)$:

$$\prod_{i=0}^{N-1} \frac{z_{t_{i+1}}(y)}{z_{t_i}(y)} = \frac{z_{t_1}(y)}{z_{t_0}(y)} \frac{z_{t_2}(y)}{z_{t_1}(y)} \cdots \frac{z_{t_N}(y)}{z_{t_{N-1}}(y)} = \frac{z_{t_N}(y)}{z_{t_0}(y)} = \frac{p(y)}{1}. \qquad (5)$$

This is known as the stepping-stone algorithm (Xie et al., 2011), and in SMC the number of levels and their temperatures can be chosen adaptively (Catanach & Beck, 2018). The tempering gradually guides the samples to areas of high posterior density, and leads to some of the most accurate estimates of $p(y)$ (Fourment et al., 2020) via the following estimator:

$$\hat{p}(y) = \prod_{i=0}^{N-1} \frac{1}{N} \sum_{j=1}^{n} p(y \mid \theta_{i,j})^{\Delta t_i}, \qquad \text{where } \{\theta_{i,j}\}_{j=1}^{n} \sim p_{t_i} \qquad (6)$$

## 4 Reverse Annealed Sequential Monte Carlo

### 4.1 Motivation: Balancing Variances

In BOED we are not interested in solely estimating $p(y)$; we are interested in EIG($d$), for which the individual IG($y$) will depend on their own $p(y)$. Hence, for an estimate of the EIG there are two

sources of variance. The first is variance in $\mathrm{IG}(y)$ across the datasets $y$, and the second is variance from its estimate $\hat{\mathrm{IG}}(y)$ for a given $y$, which we denote $\mathrm{Var}(\hat{\mathrm{IG}}(y) \mid y)$. If we assume $\mathrm{Var}(\hat{\mathrm{IG}}(y) \mid y)$ is constant across $y$ and $\hat{\mathrm{IG}}$ is unbiased, we can formalize this through the law of total variance:

$$\mathrm{Var}(\hat{\mathrm{IG}}(y)) = \mathbb{E}[\mathrm{Var}(\hat{\mathrm{IG}}(y) \mid y)] + \mathrm{Var}(\mathbb{E}[\hat{\mathrm{IG}}(y) \mid y]) = \mathrm{Var}(\hat{\mathrm{IG}}(y) \mid y) + \mathrm{Var}(\mathrm{IG}(y)) \qquad (7)$$

Our experiments find that SMC can be overly precise in the sense that $\mathrm{Var}(\hat{\mathrm{IG}}(y) \mid y) \ll \mathrm{Var}(\mathrm{IG}(y))$. Significant cost is spent on achieving excellent information gain estimates for each dataset, but this precision is unwarranted in light of the overall variance across datasets. As an example, we demonstrate this on the coupled spring-mass system of Sec. 5.1. For a fixed design, we draw 100 different $y$ using different $\theta$, and for each we compute 30 estimates $\hat{\mathrm{IG}}(y)$ using an SMC algorithm (Catanach & Beck, 2018) with 250 particles. The results are summarized in Table 1a, where we find that the variance of $\hat{\mathrm{IG}}(y)$ is two orders of magnitude less than $\mathrm{Var}(\mathrm{IG}(y)) = 22.4$.

| Dataset | $\mathrm{IG}(y_i)$ | $\mathrm{Var}\big(\hat{\mathrm{IG}}(y_i) \mid y_i\big)$ |
|---------|---------|---------|
| $y_1$ | 20.71 | 0.059 |
| $y_{51}$ | 11.40 | 0.039 |
| $y_{60}$ | 19.24 | 0.093 |
| $y_{71}$ | 23.39 | 0.110 |
| $y_{100}$ | 11.57 | 0.021 |

| # Particles | $\mathrm{IG}(y_1)$ | $\mathrm{Var}(\hat{\mathrm{IG}}(y_i) \mid y_i)$ |
|---------|---------|---------|
| 250 | 20.210 | 0.081 |
| 150 | 20.377 | 0.133 |
| 50 | 20.513 | 0.570 |
| 20 | 20.399 | 1.384 |
| 10 | 24.802 | 25.123 |

(a)          (b)

Table 1: (a) Selected variances of thirty SMC estimates of IG (250 particles). The variance of IG across all 100 datasets is 22.4. Full results in Figure 10 (appendix). (b) Comparison of $\mathrm{Var}(\hat{\mathrm{IG}}(y_1) \mid y_1)$ for different numbers of SMC particles.

We modulate the variance of the forward SMC estimator by reducing the number of particles. Shown in Table 1b, we find that reducing particles from 250 to 20 still yields estimators with an order of magnitude less variance than $\mathrm{Var}(\mathrm{IG}(y))$. Only when we get to 10 particles does $\mathrm{Var}(\hat{\mathrm{IG}}(y)) \approx \mathrm{Var}(\mathrm{IG}(y))$. These few-particle SMC estimators are no longer useful for estimating individual $\mathrm{IG}(y)$ due to their variance and thus could not be applied in other contexts–their utility is unique to calculating EIGs.

**Single particle estimates** We now consider the extreme case of using only a single particle. This seems ill-advised, however failure is a consequence of poor MCMC mixing, not failure of estimation using few samples. To show this, we consider whether we can use a single sample from a well-mixed Markov chain at each temperature. To ensure proper mixing we run forward SMC using 1000 particles, but estimate IG using only a single random particle from each temperature. This results in a variance of 14.903, which is roughly half $\mathrm{Var}(\mathrm{IG}(y))$. This demonstrates that a single particle IG estimate, from a well-mixed chain, is feasible for experimental design. To obtain such a result we temper backwards, from the posterior to the prior distribution, which we discuss next.

### 4.2   Reverse-annealed SMC Algorithm

Now we make use of a second unique feature of BOED: datasets are simulated when computing the EIG. This means that we have access to the true underlying parameter $\theta^*$. Crucially, we can treat each joint sample $(y, \theta^*)$ as being generated first by drawing $y$ from its marginal, and then $\theta^*$ from the posterior. This gives us one *free* draw from the posterior, and so we can start SMC sampling from $\theta^*$ and consider sampling a sequence of tempered distributions beginning from the posterior and going to the prior, taking a single sample at each level. This simplifies sampling as we start with a sample from the most difficult distribution and decrease temperature over time, preventing degeneracy issues where our samples end up stuck in low likelihood regions.

We call this scheme *reverse-annealed SMC* (RA-SMC). Since we will be increasing the variance in our estimates, we use the thermodynamic integral (Gelman & Meng, 1998) to help reduce bias by directly targeting $\log p(y)$:

$$\log p(y) = \int_0^1 \int_\theta \log p(y \mid \theta) \frac{p(y \mid \theta)^t p(\theta)}{z_t(y)} \, \mathrm{d}\theta \, \mathrm{d}t. \qquad (8)$$

A proof of this identity is provided in Appendix A.1.

---

**Algorithm 1** Reverse annealed sequential Monte Carlo

---

**Require:** Dataset $(\theta^*, y)$, Temperatures $t = \{t_0, \ldots, t_N\}$
**Ensure:** Output $\hat{\text{IG}}(y)$
 1: Initialize $\theta \leftarrow \theta^*$, $i \leftarrow N - 1$, $l \leftarrow Array[\log p(y \mid \theta^*)]$
 2: **while** $i \geq 0$ **do**
 3:     $\theta \leftarrow MCMC(\theta, p_{t_i})$                                      { Run MCMC on $p_{t_i}$ starting at $\theta$}
 4:     $i \leftarrow i - 1$
 5:     $l$.append($\log p(y \mid \theta)$)
 6: **end while**
 7: **return** $\log p(y \mid \theta^*) - Simpson(t, l)$      { Use Simpson's rule for thermodynamic integral.}

---

The temperature sequence provides a discretization of $t$ on $[0, 1]$, and we then use Simpson's rule (Young & Gregory, 2012) to approximate the outer integral (Calderhead & Girolami, 2009). At each tempering level, the particle is advanced via MCMC runs targeting $p_{t_i}(\theta \mid y)$, which is then used in evaluating the thermodynamic integral. By interchanging the order of integration, we can also view our estimator as first doing numerical integration with Simpson's rule using a single draw at each temperature, and then averaging over all the integrals. This enables us to estimate Monte Carlo standard errors by computing the variance across the integrals. The full RA-SMC estimator is thus defined as follows.

**Definition 4.1.** *For a fixed $y$ and design $d$, define*

$$\hat{\ell}_i^{(M)}(y, d) = \log p(y \mid \theta_i^{(M)}, d)$$

*and*

$$\widehat{\log p}_{N,M}(y, d) = S_N(\hat{\ell}_0^{(M)}(y, d), \hat{\ell}_1^{(M)}(y, d), \ldots, \hat{\ell}_N^{(M)}(y, d)).$$

*Here, $\theta_i^{(M)}$ denotes a sample drawn from an $M$-step MCMC sampler targeting $p_{t_i}$ as its stationary distribution, and $S_N$ is the composite 1/3 Simpson's rule estimator of the input integral evaluated at $N + 1$ temperature levels $t_i = (i/N)$, $i, = 0, 1, \ldots, N$. Then the* RA-SMC *estimator for design $d$ is:*

$$\widehat{EIG}(d) = \frac{1}{K} \sum_{k=1}^{K} \log p(y_k \mid \theta_k) - \widehat{\log p}_{N,M}(y_k, d), \qquad \text{where } \{(y_k, d_k)\}_{k=1}^{K} \overset{iid}{\sim} p(y, \theta)$$

We now establish a main property of RA-SMC–it is an asymptotically unbiased estimator of EIG:

**Lemma 4.2.** *Define the RA-SMC estimator as in Definition 4.1. If the MCMC sampler used to draw each $\theta_i^{(M)}$ defines an ergodic Markov chain with stationary distribution $p_{t_i}$, and if $\mathbb{E}_{p_t}[\log p(y \mid \theta)]$ is four times continuously-differentiable for all $t \in [0, 1]$, then the RA-SMC estimator is asymptotically unbiased.*

The proof of Lemma 4.2, which can be found in full in Appendix A.1, relies on the asymptotic convergence of the composite Simpson's rule estimator as well as the convergence to the correct stationary distribution of a valid MCMC sampler.

The thermodynamic integral approach is also valid for forward SMC, but the number of tempering levels needed for accurate integration tends to be higher than the number of tempering levels needed for good SMC estimates, and the bias is less of an issue when the estimate of $p(y)$ is precise. The full algorithm is presented in Algorithm 1, and despite only using a single sample at each iteration, we will see in Section 5 that its accuracy is comparable to forward SMC with far more particles.

## 5 Experimental Results

We compare our algorithm to the forward SMC algorithm of Catanach & Beck (2018), which adaptively chooses the tempering levels and MCMC iterations based on effective sample size calculations. Different numbers of particles are also considered to illustrate the discussion in Section 4.1, with a 250 particle run considered the gold standard against which we compare. For our backward estimator, we use a fixed tempering sequence based on Calderhead & Girolami (2009), with $N = 100$ levels and temperatures $t_i = (i/N)^5$ (see Corollary A.1). For the MCMC kernel we use a simple random walk

Metropolis, where the proposal standard deviation is adapted based on the previous temperature's acceptance rate using the feedback controller of Catanach (2017) to target an ideal acceptance rate of 0.234 (Gelman et al., 1997). Based on an analysis shown in Appendix A.2.1, we fix the number of tempering levels at 100 and only adjust the number of MCMC iterations per temperature for simplicity. We implement an early stopping criterion for MCMC at each level once the Spearman correlation between the starting log likelihoods and the current log likelihoods drops below 0.1. We also halve the number of steps taken once the proposal standard deviation equals the prior standard deviation, indicating that the power posterior is diffuse enough such that more iterations are not critical.

We observe better performance with fewer MCMC iterations across almost all configurations, though overestimation is now more likely. For tuning, we fix a design and run multiple MCMC iterations, stopping roughly when our estimates stabilize while accounting for Monte Carlo standard error, which ended up being 60 iterations. Any initial SMC runs used to determine whether this sampler is viable as mentioned in Section 4.1 can also be used to inform tuning.

## 5.1 Coupled spring-mass system

We first explore a coupled spring-mass system depicted in Fig. 1. Two masses, $m_1$ and $m_2$, are on a surface with respective friction coefficients $b_1$ and $b_2$ and joined by a spring with spring constant $k_2$. The first mass then joined to a fixed point by a second spring with spring constant $k_1$. The springs are assumed to have length 0 when no forces are applied to them, and the starting positions and velocities of the masses are set to 0. Each of the two masses, their friction coefficients, and the two spring constants are

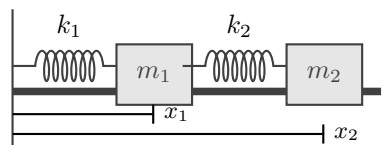

Figure 1: Coupled spring-mass system. Each of the masses also has an friction coefficient $b_1$ and $b_2$.

considered unknown parameters, so the posterior is 6D: $(m_1, m_2, b_1, b_2, k_1, k_2)$. All parameters have log-normal priors with mean 0 and SD 1. The experiment imparts a damped oscillating force on the system representing a vibration, given by:

$$x_1' = v_1, \quad x_2' = v_2, \quad v_1' = \frac{-b_1 v_1 - (k_1 + k_2)x_1 + k2 x_2}{m_1}, \quad v_2' = \frac{-b_2 v_2 + k_2(x_1 - x_2) + f(\gamma, t)}{m_2}$$

where $f(\gamma, t) = 5\sin(\gamma t)\exp(-t/5)$ is the forcing function, $x_i$ is the position of the $i$th mass, and $v_i$ is the velocity of the $i$th mass. We observe a noisy position $x_1$ at 100 equally spaced time points. The noise is Gaussian with mean 0 and SD 0.025, with observed data $y = \{y_1, \ldots, y_{100}\}$ and,

$$y_t = x_{1t} + \epsilon_t, \quad \epsilon_t \overset{iid}{\sim} N(0, 0.025^2), \quad t = 1, \ldots, 100$$

where $x_{1t}$ is the position of the first mass at time $t$. Fig. 2 shows three randomly generated datasets. We consider ten equally spaced designs $\gamma$ between 0 and 2 of the form $0.2j, j = 1, \ldots, 10$. By

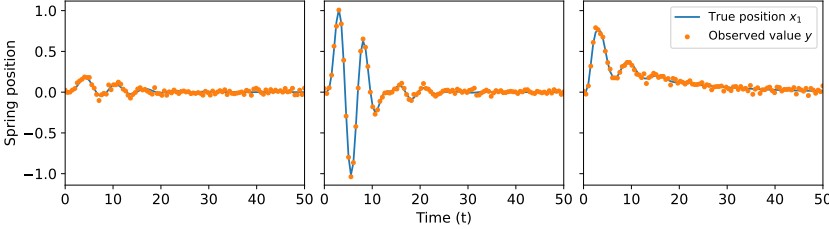

Figure 2: Three examples of observed data $y$ in orange, along with the true signal $x$ in blue for the spring-mass model. The true signal corresponds to the position of the first mass $m_1$ between time $t = 0$ and $t = 50$, and is observed over 100 equally-spaced time points.

observing only one mass, we induce multimodality and curvature in the posterior distribution. Fig. 3 shows four dimensions of an example posterior. 1000 datasets are drawn for each method, with performance measured by the number of likelihood evaluations required and how well the final EIG values correspond to those of an expensive forward SMC run. Likelihood evaluations are the dominant cost and are therefore used as a hardware agnostic metric for computational efficiency.

**Magnetic reverse-annealed SMC steps** To aid MCMC mixing we use knowledge of the target distribution (the prior), when tempering backwards, to inform the initialization at each level. The idea is that the next temperature level should place more mass towards the prior. Hence, instead of starting $\theta$ at the last position of the previous level, we nudge the starting point of the MCMC towards the prior mean. Between levels, we take up to a tenth of the planned MCMC iterations toward the prior mean using the proposal standard deviation as the step size, stopping if the log likelihood at any new step is less than a tenth of the starting log likelihood. We note that this is an optional enhancement, rather than a core component of the method.

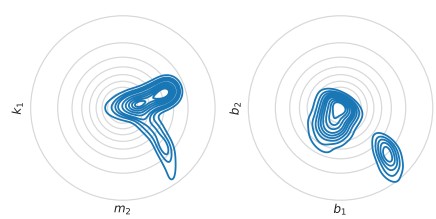

Figure 3: Example spring-mass posterior for four of the parameters based on kernel density estimation using samples from 250-particle forward SMC, with the standard normal priors shown in grey. The posterior demonstrates multimodality and curvature.

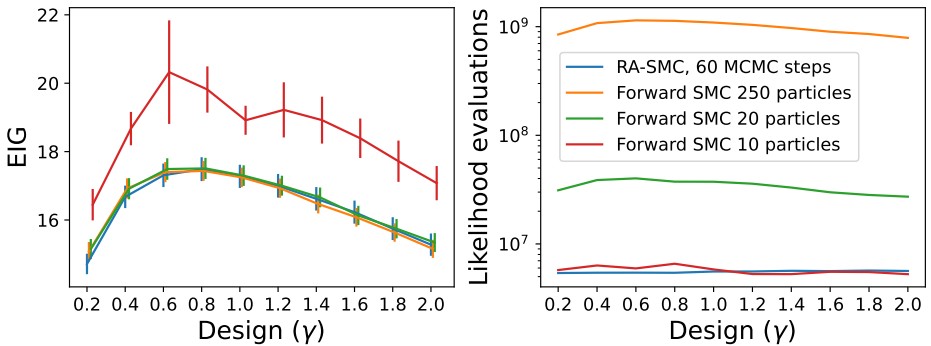

Figure 4: Left: EIG estimates across 10 designs of the spring-mass model, with vertical bars representing two Monte Carlo standard errors above and below the estimate. Right: Number of likelihood evaluations for each estimate. The adaptive nature of forward SMC results in a varying number of evaluations per design.

**Results** A comparison of the EIG curves resulting from traditional forward SMC estimators with varying numbers of particles to the backward SMC estimator's EIG curve is shown in Figure 4. By decreasing the number of forward SMC particles by an order of magnitude to 20, we gain an order of magnitude of efficiency with little loss of performance compared to the gold standard 250 particle runs.

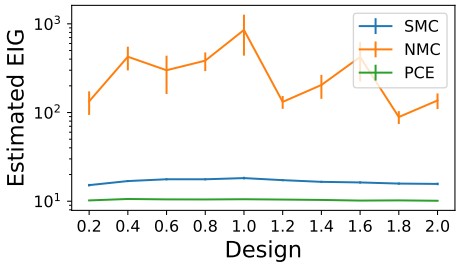

Figure 5: Estimated expected information gain (EIG) for the springmass design problem across design values from 0.2 to 2.0. Each curve corresponds to a different estimator: Forward SMC, Nested Monte Carlo (NMC), and Prior Contrastive Estimation (PCE). Error bars indicate estimator standard errors, computed across replicates. SMC estimates use 100 outer samples and 20 particles per stage; NMC and PCE estimates use 100 outer samples and 300,000 inner samples. Error bars for the SMC and PCE curves are present but too small to be visible.

At 10 particles, significant upward bias and noise occurs. Notably, the failure is catastrophic as opposed to gradual; below some threshold, mixing fails and the estimates are poor, while above that threshold the estimates are still stable. Our reverse estimator performs well with another fourfold decrease in likelihood evaluations compared to the 20 particle forward SMC.

**Other Monte Carlo Baselines** As a baseline, we also evaluated Nested Monte Carlo (NMC) and Prior Contrastive Estimation (PCE) on the spring-mass problem, allocating the same computational budget as the 20-particle forward SMC run. As shown in Figure 5, both estimators performed poorly. NMC produced inaccurate estimates with large standard errors, while PCE exhibited a strong downward bias and failed to locate the optimal design. Since the data are informative, the resulting posterior is sharply peaked. The prior-based sampling of these methods is inadequate to explore this region, which leads to unstable and biased evidence estimation.

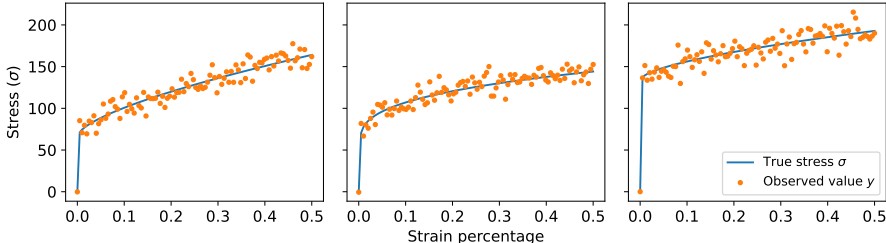

Figure 6: Three examples of observed data $y$ in orange, along with the true signal $\sigma$ in blue for the Johnson-Cook model. The true signal corresponds to the output stress of a material for strain percentage $\varepsilon$ between 0 and 0.5, observed at 100 equally spaced values.

## 5.2 Johnson-Cook model

We run a Johnson-Cook model of a stress-strain curve for a hypothetical material under plastic deformation. The data follows the following model, where the experiment involves observing the stress at 100 equally spaced strain percentages $\varepsilon$ from 0 to 0.5, with varying measurement noise dependent on whether the material is in the elastic or plastic phase:

$$
y = \begin{cases} E\varepsilon + \delta_e & \text{if } \varepsilon E < A \\ \left(A + B\left(\varepsilon - \dfrac{A}{E}\right)^n\right) \times (1 + C\log(\dot{\varepsilon})) \times \left(1 - \left(\dfrac{T - 293}{775 - 293}\right)^m\right) + \delta_p & \text{otherwise,} \end{cases}
$$

where $\delta_e \overset{iid}{\sim} N(0, 1)$ and $\delta_p \overset{iid}{\sim} N(0, 10)$. We consider the strain rate $\dot{\varepsilon}$ and temperature $T$ as experimental variables. Other parameters are unknown material constants with the following priors:

$$
E \sim \mathcal{N}(73000, 10000^2), \quad A \sim \mathcal{N}(350, 100^2), \quad B \sim \mathcal{N}(650, 200^2),
$$
$$
C \sim \text{Beta}(2, 10), \quad n \sim \text{Beta}(2, 5), \quad m \sim \text{Beta}(2, 5).
$$

Figure 6 shows examples of three randomly generated datasets for $\dot{\varepsilon} = 0.1$ and $T = 300$. For simplicity, we evaluate five strain rates $\{0.001, 0.005, 0.01, 0.05, 0.1\}$ for the same temperature 300, and then multiple temperatures $\{300, 400, 500, 600, 700\}$ for the same strain rate 0.1. For the MCMC kernel, we use a proposal standard deviation equal to the prior standard deviations scaled to target the 0.234 acceptance rate, though no adaptation of the proposal to the prior is done since the prior is not multivariate Gaussian. Similarly to the coupled spring-mass model in Section 5.1, we use 1000 datasets with 60 MCMC steps as well as the magnetic SMC steps.

**Results** Figure 7 displays the EIG curves resulting from forward SMC estimators with different population sizes compared to the EIG curve generated by the reverse-annealed SMC estimator with varying MCMC iterations when using both strain rate and temperature as the design variable. Similarly to the spring-mass model, we see that we can decrease the number of particles to 20 without significant impact to the EIG curve in both cases. We also find that the reverse-annealed SMC estimator with 60 MCMC steps achieves good results. Moreover, the computation budget for either configuration is roughly equivalent to forward SMC with 10 particles, while yielding substantially more accurate EIG estimates.

Unlike with the spring-mass model, the reverse-annealed SMC estimator with 10 MCMC steps is not as heavily biased downward, providing a better estimator than the 10-sample forward SMC estimator and requiring an order of magnitude fewer likelihood evaluations. When using temperature as the design variable, the EIG curve from the reverse-annealed SMC estimator matches the "true" EIG curve even better than when using strain rate as the design. Unsurprisingly, we still see a higher standard error than reverse-annealed SMC with 60 MCMC steps, although the standard error is lower in the temperature design setting than in the strain-rate design setting.

## 5.3 Source Finding

Finally, we apply our reverse-annealed SMC estimator to the source-localization problem described in Foster et al. (2021), in which two hidden sources emit a signal whose intensity decays according to the inverse-square law. At each step, a sensor placed at a design location $\xi$, records a noisy measurement of the aggregate signal intensity. The objective is to sequentially choose sensor placements $\{\xi_t\}_{t=1}^{T}$

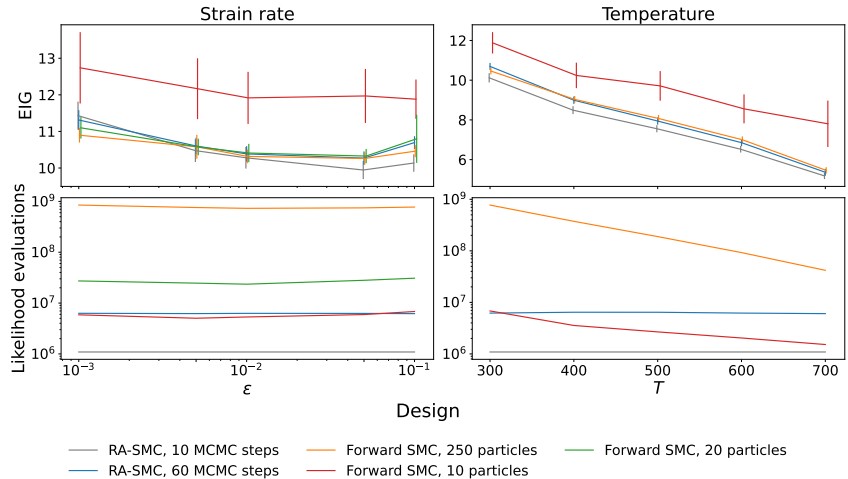

Figure 7: EIG and computational cost results for different under two different choices of design variable for the Johnson-Cook model. Left column: EIG and computational cost for five choices of strain rate at a fixed temperature 300. Reverse-annealed SMC yields accurate EIG curves with orders of magnitude fewer likelihod evaluation compared to forward SMC baselines. Right column: EIG and computational cost results for five choices of temperature at a fixed strain rate 0.1. Here, forward SMC is able to effectively adapt the number of iterations required at the higher temperatures. Error bars indicate two Monte Carlo standard errors.

to efficiently infer the unknown source coordinates $\theta$ from the noisy observations $\{y_t\}_{t=1}^T$. The noiseless observation model is:

$$\mu(\theta, \xi) = b + \sum_{k=1}^{K} \frac{1}{m + \|\theta_k + \xi\|^2},$$

where $b, m > 0$ are constants that control background and maximum signal, respectively. We observe the log intensity, and use the following prior and likelihood models:

$$\theta_k \overset{\text{iid}}{\sim} \mathcal{N}(0, I_2), \quad \log y \mid \theta, \xi \sim \mathcal{N}(\log \mu(\theta, \xi), \sigma).$$

Each observation is chosen by selecting: $\xi_t = \arg\max_\xi \text{EIG}(\xi \mid y_{1:t-1}, \xi_{1:t-1})$. Once $\xi_t$ is chosen, the observer records $y_t$, the posterior is updated to $p(\theta \mid y_{1:t}, \xi_{1:t})$, and the design-measure-update cycle is repeated until a terminal time $T$ is reached.

In our experiments we choose $m = 10^{-4}$ and $b = 10^{-1}$, matching the values used in Foster et al. (2021), and we restrict observation locations to a $20 \times 20$ grid covering the square domain $[-3, 3] \times [-3, 3]$. We perform $T = 15$ steps of sequential optimization using RA-SMC and forward SMC with 50 and 100 particles.

**Results**   Figure 8 shows the results of the source-finding problem using reverse-annealed SMC and forward SMC with 50 and 100 particles. We find that reverse-annealed SMC not only produces tightly clustered sensor placements around the true sources, but does so with substantially less computation. Over 15 measurements, reverse-annealed SMC required $8.4 \times 10^9$ likelihood evaluations, whereas forward SMC consumed about $3.7 \times 10^{10}$ and $4.8 \times 10^{10}$ evaluations for 50 and 100 particles, respectively.

Despite this reduction in evaluation cost, reverse-annealed SMC yields better convergence than 50-particle forward SMC and similar performance to 100-particle forward SMC. Figure 9 shows the KL divergence between the prior and the posterior at each iteration of the sequential design process. We see that RA-SMC performs similarly to 100-particle forward SMC, while 50-particle forward SMC gives erratic results.

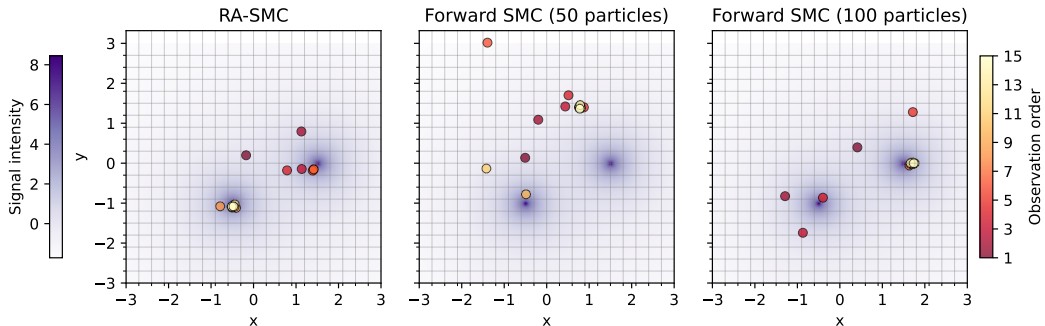

Figure 8: Side-by-side comparison of sequential sensor placements for source localization using reverse-annealed SMC (left), forward SMC with 50 particles (middle) and forward SMC with 100 particles (right). The background heatmap indicates the log-signal intensity over the domain, and each marker shows a sensor location colored by its observation order (dark to light). Reverse-annealed SMC and 100-particle SMC rapidly concentrate measurements near the true source positions, whereas 50-particle forward SMC exhibits a more dispersed sampling pattern. The likelihood evaluations were $8.4 \times 10^9$, $3.7 \times 10^{10}$, and $4.8 \times 10^{10}$, respectively.

## 5.4 Discussion and Limitations

The single-particle, reverse trajectory of RA-SMC has key limitations. It complicates adaptive MCMC proposals and tempering schedules, which often require population statistics. Poor MCMC mixing when annealing from a peaked posterior toward the flatter prior can yield conservative IG estimates. Determining its suitability for a problem is currently ad hoc, requiring preliminary runs to find efficiency gains. A robust diagnostic for this would therefore be valuable.

On the other hand, because RA-SMC initializes from a true posterior sample, it may be less susceptible to the particle degeneracy issues that hinder traditional multi-particle SMC methods in high dimensions. Future research could explore developing diagnostics to assess estimator suitability, incorporating more sophisticated MCMC kernels (e.g., Hamiltonian Monte Carlo (Neal, 2012) or the No-U-Turn sampler (Hoffman et al., 2014)), studying the method's scalability in high dimensions, devising adaptive strategies using multiple particles to improve both MCMC mixing and the selection of tempering levels, extending the framework to real-world data by initializing the reverse trajectory with approximate posterior draws from likelihood-free methods, and conducting a more extensive comparison to other estimators for BOED.

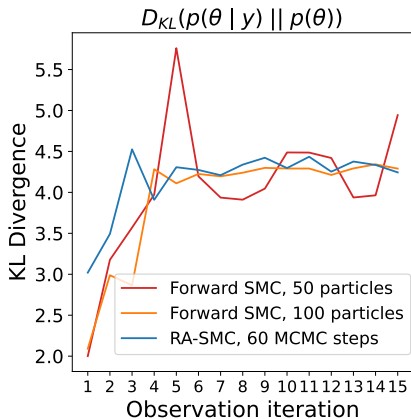

Figure 9: Evolution of the KL divergence over 15 sequential observations for forward SMC with 50 particles (red), forward SMC with 100 particles (orange), and reverse-annealed SMC with 60 MCMC steps (blue). RA-SMC converges as smoothly and rapidly as 100-particle forward SMC, while 50-particle forward SMC yields more erratic KL divergence estimates.

## Acknowledgments and Disclosure of Funding

Sandia National Laboratories (SNL) is a multimission laboratory managed and operated by National Technology & Engineering Solutions of Sandia, LLC, a wholly owned subsidiary of Honeywell International Inc., for the U.S. Department of Energy's National Nuclear Security Administration under contract DE-NA0003525. SAND#2025-13387C. Initial research was supported by SNL Laboratory Directed Research and Development. This paper describes objective technical results and analysis. Any subjective views or opinions that might be expressed in the paper do not necessarily represent the views of the U.S. Department of Energy or the United States Government.

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

# A Appendix

## A.1 Proofs and additional theory

**Proof of Lemma 4.2.**

*Proof.* By the linearity of expectations,

$$\mathbb{E}[\widehat{\log p}_{N,M}(y,d)] = S_N(\hat{\ell}_0^{(M)}(y,d), \hat{\ell}_1^{(M)}(y,d), \ldots, \hat{\ell}_N^{(M)}(y,d))$$

$$= \mathbb{E}\left[\frac{\Delta t}{3}\left(\hat{\ell}_0^{(M)}(d) + \hat{\ell}_N^{(M)}(d) + 4\sum_{\substack{i=1,\\ i\text{ odd}}}^{N-1} \hat{\ell}_i^{(M)}(d) + 2\sum_{\substack{i=2,\\ i\text{ even}}}^{N-2} \hat{\ell}_i^{(M)}(d)\right)\right]$$

$$= \frac{\Delta t}{3}\left(\mathbb{E}[\hat{\ell}_0^{(M)}(d)] + \mathbb{E}[\hat{\ell}_N^{(M)}(d)] + 4\sum_{\substack{i=1,\\ i\text{ odd}}}^{N-1} \mathbb{E}[\hat{\ell}_i^{(M)}(d)] + 2\sum_{\substack{i=2,\\ i\text{ even}}}^{N-2} \mathbb{E}[\hat{\ell}_i^{(M)}(d)]\right).$$

For each $i$, we have

$$\lim_{M\to\infty} \mathbb{E}[\hat{\ell}_i^{(M)}(d)] = \mathbb{E}_{p_{t_i}}[\log p(y \mid \theta, d)] := \ell(t_i, y, d),$$

so

$$\lim_{M\to\infty} \mathbb{E}[\hat{p}_{N,M}(y,d)] = \frac{\Delta t}{3}\left(\ell(t_0, y, d) + \ell(t_N, y, d) + 4\sum_{\substack{i=1,\\ i\text{ odd}}}^{N-1} \ell(t_i, y, d) + 2\sum_{\substack{i=2,\\ i\text{ even}}}^{N-2} \ell(t_i, y, d)\right)$$

$$:= Q_N.$$

When the fourth derivative is bounded, the composite Simpson rule satisfies the remainder bound:

$$\left|Q_N - \int_0^1 \ell(t, y, d)\, dt\right| = O(N^{-4}),$$

so

$$\lim_{N\to\infty} Q_N = \int_0^1 \ell(t, y, d)\, dt = \int_0^1 \mathbb{E}_{\theta\sim p_t}[\log p(y \mid \theta, d)]\, dt = \log p(y).$$

Hence,

$$\lim_{N,M\to\infty} \hat{p}_{N,M}(y,d) = \log p(y \mid d).$$

Finally, observe that

$$\lim_{N,M,K\to\infty} \mathbb{E}[\widehat{\text{EIG}}_{N,M,K}(d)] = \lim_{K\to\infty} \mathbb{E}\left[\frac{1}{K}\sum_{k=1}^{K} \log p(y_k \mid \theta_k, d) - \lim_{N,M\to\infty} \hat{p}_{N,M}(y_k, d)\right]$$

$$= \lim_{k\to\infty} \mathbb{E}_{(y,\theta)\sim p(y,\theta)}\left[\log p(y \mid \theta, d) - \lim_{N,M\to\infty} \mathbb{E}[\hat{p}_{N,M}(y,d)]\right]$$

$$= \mathbb{E}_{(y,\theta)\sim p(y,\theta)}[\log p(y \mid \theta, d) - \log p(y \mid d)]$$

$$= \text{EIG}(d).$$

Thus, the RA-SMC estimator of $\text{EIG}(d)$ is asymptotically unbiased. $\square$

**Corollary A.1.** *Under the same conditions as Lemma 4.2, the RA-SMC estimator is asymptotically unbiased for geometrically spaced temperature levels $t_j = (u_j)^c$, where $u_j = j/N$ are uniformly spaced points in $[0, 1]$ and $c > 1$ is a constant.*

*Proof.* We use a standard change of variables. The integral for $\log p(y \mid d)$ is transformed:

$$\int_0^1 \underbrace{\ell(u^c, y, d) c u^{c-1}}_{:=g(u,y,d)} = \int_0^1 \ell(t, y, d) dt = \log p(y \mid d).$$

The estimator for $\log p(y \mid d)$ is then constructed by applying the composite Simpson's rule $S_N(\cdot)$ (with uniform step $\Delta u = 1/N$) to MCMC-based estimates $\hat{g}_j^{(M)}(y, d)$ of the transformed integrand $g(u_j; y, d)$. Specifically,

$$\hat{g}_j^{(M)} = c(u_j)^{c-1}\hat{\ell}_j^{(M)}(y, d),$$

where $\hat{\ell}_j^{(M)}(y, d)$ is the MCMC estimate of $\ell((u_j)^c, y, d)$.

The asymptotic unbiasedness argument then applies directly to this estimation of $\int_0^1 g(u, y, d)du$ because of the following two facts. First, the transformed integrand $g(u, y, d)$ is four-times continuously differentiable on $[0, 1]$. Second, the MCMC estimates $\hat{g}_j^{(M)}(y, d)$ are asymptotically unbiased in expectation for $g(u_j, y, d)$ as $M \to \infty$: observe that

$$
\begin{aligned}
\lim_{M \to \infty} \mathbb{E}[\hat{g}_j^{(M)}(y, d)] &= \lim_{M \to \infty} \mathbb{E}[cu^{c-1}\hat{\ell}_j^{(M)}(y, d)] \\
&= \mathbb{E}[cu^{c-1}\ell(t_i, y, d)] \\
&= \mathbb{E}[g(u_i, y, d)].
\end{aligned}
$$

Under these conditions on the transformed problem, the same $O(N^{-4})$ convergence for the Simpson's rule discretization error (for $g(u)$) and the vanishing MCMC error ensure the asymptotic unbiasedness for estimating $\log p(y \mid d)$ with geometrically spaced temperatures. $\qquad\square$

**Proof of Equation 8.**

*Proof.* Observe that

$$
\begin{aligned}
\log p(y) &= \log(z_1(y)) - \log(z_0(y)) \\
&= \int_0^1 \frac{d}{dt} \log(z_t(y)) \, dt \\
&= \int_0^1 \frac{1}{z_t(y)} \frac{d}{dt} z_t(y) \, dt \\
&= \int_0^1 \frac{1}{z_t(y)} \frac{d}{dt} \int_\theta p(y \mid \theta)^t p(\theta) \, d\theta \, dt \\
&= \int_0^1 \frac{1}{z_t(y)} \int_\theta \frac{d}{dt} p(y \mid \theta)^t p(\theta) \, d\theta \, dt \\
&= \int_0^1 \frac{1}{z_t(y)} \int_\theta \log p(y \mid \theta) p(y \mid \theta)^t p(\theta) \, d\theta \, dt \\
&= \int_0^1 \int_\theta \log p(y \mid \theta) \frac{p(y \mid \theta)^t p(\theta)}{z_t(y)} \, d\theta \, dt,
\end{aligned}
$$

as desired. $\qquad\square$

## A.2  Additional Results

### A.2.1  Inner-loop variance vs. outer-loop variance

To validate that $\mathrm{Var}(\hat{\mathrm{IG}}(y) \mid y) \ll \mathrm{Var}(\mathrm{IG}(y))$, we draw 100 datasets $y_i$, $i = 1, \ldots, 100$ from the spring mass observation model in Sec 5.1. For each of these datasets, we repeatedly compute $\mathrm{IG}(y_i)$ thirty times using forward SMC with 250 particles. Then, we compute $\mathrm{Var}(\hat{\mathrm{IG}}(y) \mid y = y_i)$ across the thirty replicates of $\mathrm{IG}(y_i)$ and compare the estimator variance for a given dataset to that dataset's value of $\mathrm{IG}(y)$. Figure 10 displays this comparison for all 100 datasets. It is clear that the inner-loop estimator variance $\mathrm{Var}(\hat{\mathrm{IG}}(y) \mid y)$ is much smaller than $IG(y)$.

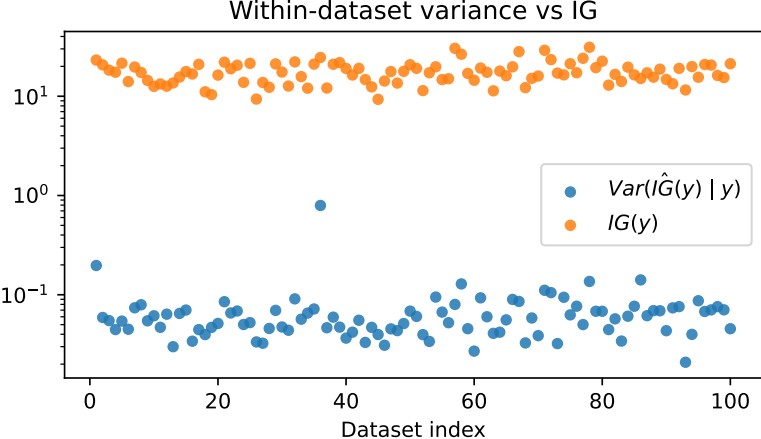

Figure 10: A comparison between $\text{Var}(\hat{IG}(y) \mid y)$, the estimator variance from computing $\text{IG}(y)$ for a single dataset, and the value of $\text{IG}(y)$. We see that $\text{Var}(\hat{IG}(y) \mid y)$ is on average two orders of magnitude smaller than $\text{IG}(y)$.

### A.2.2 Choosing the tempering levels

There is a balance between the number of temperature levels and number of MCMC iterations because more temperatures means that the difference between subsequent distributions is smaller; fewer MCMC iterations should then be required to achieve convergence. Assuming the number of levels is sufficiently large to calculate the thermodynamic integral, it is unclear whether, for example, 100 levels of 100 iterations is generally better than 200 levels of 50 iterations.

Figure 11 demonstrates how the absolute error changes for the model in Section 5.1 as we vary these parameters when compared to a gold standard forward SMC run, and it appears that there is no significant difference past a certain threshold.

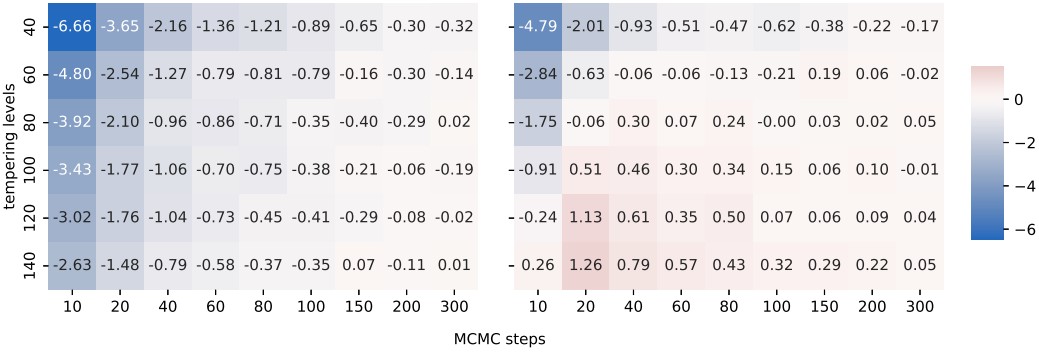

Figure 11: Left: Absolute error of EIG computed with the standard reverse-annealed SMC, with downward bias trend clearly visible when levels or steps are too low. Right: Error of magnetized reverse-annealed SMC. Values are in comparison to a long 250 particle forward SMC estimate, with Monte Carlo standard errors around 0.3.

### A.3 Basic reverse-annealed SMC

We also run a basic version of our estimator with only 10 MCMC iterations for each temperature and no further adaptivity, shown in Figure 12. As expected, this yields underestimates of the EIG. However, the resulting EIG curve can still be used for BOED as the bias is sufficiently similar across designs. The curve is also still relatively smooth due to the lack of degenerate samples increasing the variance. Even if accurate EIG estimates are required, this could be used to enable a coarse first pass to narrow the search space of designs.

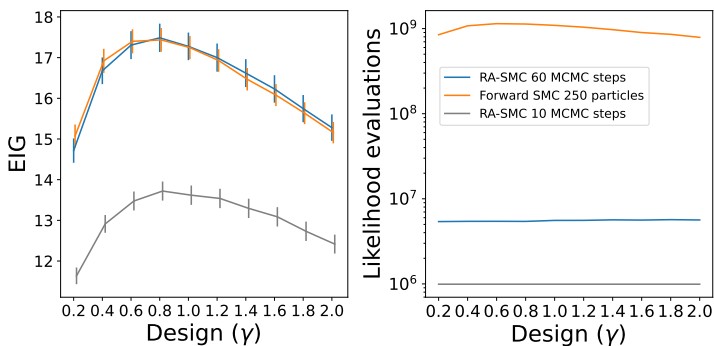

Figure 12: Comparison of results using 10 iterations of MCMC per temperature for reverse-annealed SMC on the spring-mass model. The EIGs are clearly underestimated but still form a usable curve for BOED.

One caveat is that the reverse-annealed SMC has around a 15% higher Monte Carlo standard error, 0.167 vs. 0.145 for forward SMC. This is not surprising since we are increasing $\mathrm{Var}(\hat{\mathrm{IG}}(y) \mid y)$, and so it is reasonable to say that cost of obtaining EIG estimates with the same precision as forward SMC in this case would require about 33% more outer-loop samples than were run in our simulations. However, our main takeaways do not change, as an additional 33% cost is small compared to reduction in inner-loop likelihood evaluations.

