# OpenReview forum: "Reverse-Annealed Sequential Monte Carlo for Efficient Bayesian Optimal Experiment Design"
_NeurIPS.cc/2025/Conference — NeurIPS 2025 poster_

### Official Review · Reviewer_Z8Ri · 2025-06-30

**Clarity:** 4
**Significance:** 2
**Originality:** 3
**Rating:** 5
**Confidence:** 5

**Summary:**

- EIG estimation is all about computing $E_{y\sim p(y)}[\log p(y)]$
- The variance over $y$ can be as significant as the variance/error coming from the estimate of the partition function
- A small number of inner particles can be sufficient
- In fact we can use 1 particle with an improved backwards SMC method, by observing that we get exactly 1 free sample of the posterior since EIG estimation uses data simulated from the current prior
- the resulting method gives good EIG estimates at a lower likelihood evaluation budget compared to earlier SMC-based approaches

**Questions:**

- what about EIG gradients w.r.t. $d$? This would be an interesting angle to study as I don't believe it has been attempted using SMC before (though I could be wrong)
 - "Magnetic reverse-annealed SMC steps" this sounds like an interesting addition, but could that not be a general part of the algorithm? Were the results for the mass-spring system significantly worse without this amendment?

**Ethical Concerns:**

["NO or VERY MINOR ethics concerns only"]

**Final Justification:**

Thank you to the authors for their thorough response to my questions. My score is maintained as "Accept"

**Limitations:**

Yes

**Paper Formatting Concerns:**

No concerns

**Quality:**

4

**Strengths And Weaknesses:**

# Strengths
 - the paper reads very well, it clearly lays out the overall story in a logical manner
 - the idea leads to a more efficient algorithm
 - I believe this approach to be novel
 - theoretical analysis in the form of an asymptotic unbiasedness result is provided
 - clear demonstration in the experiments that this approach requires fewer likelihood evaluations to achieve good quality results compared to some earlier SMC-based algorithms for EIG optimization

# Weaknesses
The biggest shortfalls are things that weren't in the paper
 - my question with SMC approaches for BOED is always how they will scale with dimensionality. As far as I can see, scaling with dimensionality of $\theta$ was not studied here. I think this could be worthy of investigation. In fact, the fact that you are utilizing a single particle initialized from your true theta sample gives me some hope that this might be more scalable than other SMC approaches. I think adding this would have made the paper even better
 - you only compare against other SMC approaches except in Appendix A.3 where you study NMC. I would definitely be interested to see Prior Contrastive Estimation (PCE) https://arxiv.org/pdf/1911.00294 eq (12) on the table in A.3 as well-- it can be a deceptively good baseline for virtually the same cost as NMC. Ideally it would be added to the main plots as well. My motivation to suggest this is that you make an argument that some error in $\log p(y)$ is fine because of the variance over $y$. In that case, maybe we can start to tolerate these even cruder approximations?

---

> ### Author Rebuttal · Authors · 2025-07-30
>
> We sincerely thank Reviewer Z8Ri for their thoughtful review and for the insightful questions and suggestions. We were particularly encouraged by the reviewer's optimism about our method's potential.
>
> 1. > ...how they will scale with dimensionality...was not studied here.
>
> We agree that scalability is an important question, and we thank the reviewer for raising this point. Like the reviewer, we **are optimistic about RA-SMC's potential for scalability.** Since we are initializing from a true posterior we are optimistic that we can avoid particle degeneracy issues that hinder traditional multi-particle SMC methods in high dimensions.
>
> Our primary goal in this work was to first establish the core methodology and validate its efficiency on problems known to be challenging due to complex, multimodal posteriors, rather than high dimensionality. A full investigation into dimensionality scaling is an excellent direction for a follow-up study.
>
> 2. > ...only compare against other SMC approaches ... I would definitely be interested to see Prior Contrastive Estimation (PCE) on the table in .3 as well...maybe we can start to tolerate these even cruder approximations?
>
> We thank the reviewer for this suggestion! To address it directly, we **ran a new experiment during the rebuttal period comparing our method to Prior Contrastive Estimation (PCE)** on the spring mass problem.
> Our results show that for these problems, the bias from PCE is too severe to be useful.
>
> |Design|0.2|0.4|0.6|0.8|1.0|1.2|1.4|1.6|1.8|2.0|
> |------|---|---|---|---|---|---|---|---|---|---|
> |EIG - PCE   |10.205|10.600|10.487|10.469|10.531|10.443|10.343|10.166|10.223|10.121|
> |SE - PCE |0.070 |0.061 |0.063 |0.074 |0.026 |0.044 |0.068 |0.063 |0.048 |0.077|
> |EIG - Forward SMC | 16.065 | 18.455| 20.269| 19.339|19.034|19.648|18.439|18.884|17.349|16.755|
>
> As shown in the table, PCE consistently severely underestimates the EIG and fails to identify the optimal design around $\gamma = 0.6$. While it performs better than NMC, PCE also demonstrates the need for more powerful estimators.
>
> 3. > what about EIG gradients w.r.t. $d$?
>
> We thank the reviewer for this interesting suggestion. We agree that this would be very interesting to study and to our knowledge it hasn't been studied in the context of SMC samplers before.
>
>
> 4. > Magnetic reverse-annealed SMC steps
>
> The ``magnetic'' steps represent an auxiliary enhancement we explored during development where we nudge the sampler toward the prior mean between temperature levels to potentially aid mixing.
> While we wanted to demonstrate another opportunity to take advantage of the unique reverse annealing setting to improve efficiency, we view it as separate from the core algorithm because it's not necessary for good performance (as evidenced by our source-finding experiment where we omitted it entirely) and it complicates the theoretical analysis. The spring-mass results were not significantly worse without the magnetic steps, but we did see a small improvement.
> **We will clarify in the revision that these steps are an optional enhancement rather than a core component of the method.** The fundamental RA-SMC algorithm, starting from an exact posterior sample and tempering backwards using standard MCMC, represents our primary contribution.

---

> > ### Comment · Reviewer_Z8Ri · 2025-08-04
> > **Thank you for your response**
> >
> > I would like to thank the authors for their detailed rebuttal and for taking the time to try out the PCE baseline. I do believe these results help to buttress the results of the paper and provide encouraging evidence that SMC-based methods are indeed competitive against alternative approaches.
> >
> > I do also think that some clarification around the "magnetic" part of the algorithm would help, and also that it would be beneficial to clearly highlight the point that "the spring-mass results were not significantly worse without the magnetic steps, but we did see a small improvement"
> >
> > Overall I remain very positive about this paper's inclusion in NeurIPS

---

### Official Review · Reviewer_2iaK · 2025-07-01

**Clarity:** 4
**Significance:** 4
**Originality:** 4
**Rating:** 6
**Confidence:** 4

**Summary:**

This paper introduces a novel approach called Reverse-Annealed Sequential Monte Carlo (RA-SMC) to estimate the Expected Information Gain (EIG) in Bayesian Optimal Experimental Design (BOED). The method leverages the specific characteristics of BOED to achieve efficient estimation, requiring only a single sample (particle) to provide a good approximation. The paper also provides a comprehensive theoretical foundation for the proposed method and demonstrates its efficacy on various models, showing significant computational savings compared to traditional methods.

**Questions:**

1. Could you provide a formal proof or further justification for the validity of Formula (8)? This would address any concerns about the theoretical foundation of the method.

2. While the paper demonstrates the sufficiency of using a single particle, could you explore and discuss the potential use of multiple particles (perhaps inspired by particle filters) to further enhance the performance of RA-SMC, especially for complex or multimodal posteriors?

3. Given that "Sequential Monte Carlo" is widely used in the context of filtering, would it be possible to reconsider the use of this term for the proposed method? Alternatively, can you provide a clearer distinction between your method and traditional filtering-based SMC techniques?

4. It would strengthen the paper to include a comparison with other modern methods for EIG estimation in BOED. This could help further validate the efficiency and accuracy of RA-SMC.

**Ethical Concerns:**

["NO or VERY MINOR ethics concerns only"]

**Final Justification:**

Thank authors for their thorough and thoughtful responses. I am satisfied with the clarifications provided and maintain my original score: Strong Accept.

**Limitations:**

Yes

**Paper Formatting Concerns:**

No major formatting issues were found.

**Quality:**

4

**Strengths And Weaknesses:**

Strengths:

1. The idea of using reverse annealing for EIG estimation in BOED is highly innovative. The authors effectively exploit the characteristics of BOED, such as known underlying parameters and the goal of estimating EIG, to propose a method that drastically reduces the computational cost.

2. The paper provides a solid theoretical basis for RA-SMC, demonstrating that a single particle can effectively estimate EIG. The proofs and discussions are well-detailed.

3. The method is shown to significantly reduce the number of likelihood evaluations required, offering computational efficiency without sacrificing accuracy. This is demonstrated across several models.

4. The paper is clearly written, with a logical flow from introducing the problem to detailing the RA-SMC approach and providing empirical results. The experimental setup is easy to follow, and the comparison with the forward SMC method is well-executed.

Weaknesses:

1. Eq. (8) is fundamental to the method's theoretical framework, but its validity is not immediately clear, and the authors have not provided a proof. A formal proof or additional justification for this formula would strengthen the theoretical rigor of the paper.

2. While the paper demonstrates that a single particle is sufficient for RA-SMC, there is limited discussion on the potential benefits of using multiple particles. It would be useful to explore whether using multiple particles (similar to particle filter methods) could improve performance, especially in cases with high posterior variance or complex models.

3. The term "Sequential Monte Carlo (SMC)" is widely used in various fields, particularly in filtering and dynamic systems. While this term may be appropriate within the context of your work, I would like to point out that this could lead to some confusion for readers coming from other areas. It may be helpful to clearly define the SMC terminology in the context of your work, or alternatively, consider a different name to distinguish your approach from the SMC methods for filtering used in other domains.

4. The paper only compares RA-SMC with forward SMC, but it would be more informative to include comparisons with other state-of-the-art methods for EIG estimation.

---

> ### Author Rebuttal · Authors · 2025-07-30
>
> We sincerely thank the reviewer for their positive evaluation, high confidence, and insightful feedback. We are encouraged by their assessment and appreciate the constructive suggestions for improvement. We address each point below.
>
> 1. > Could you provide a formal proof or further justification for the validity of Formula (8)?
>
> We agree that a short derivation would improve clarity.
> The identity follows from the Fundamental Theorem of Calculus, which we include below for convenience.
> \begin{aligned}
>     \log p(y)
>     &= \log p(y) - \log (1)
>     \\\\
>     &= \log (z_1(y)) - \log (z_0(y))
>     \\\\
>     &= \int_0^1 \frac{d}{dt} \log(z_t(y)) \,dt
>     \\\\
>     &= \int_0^1 \frac{1}{z_t(y)} \frac{d}{dt} z_t(y) \,dt
>     \\\\
>     &= \int_0^1 \frac{1}{z_t(y)} \frac{d}{dt} \int_{\theta}p(y \mid \theta)^t p(\theta)\,d\theta \,dt
>     \\\\
>     &= \int_0^1 \frac{1}{z_t(y)} \int_{\theta} \frac{d}{dt} p(y \mid \theta)^t p(\theta)\,d\theta \,dt
>     \\\\
>     &= \int_0^1 \frac{1}{z_t(y)} \int_{\theta} \log p(y \mid \theta) p(y \mid \theta)^t   p(\theta)\,d\theta \,dt
>     \\\\
>     &= \int_0^1 \int_{\theta} \log p(y \mid \theta) \frac{p(y \mid \theta)^t   p(\theta)}{z_t(y)}\,d\theta \,dt.
> \end{aligned}
>
> **We have added this derivation to Appendix A.1 and added a reference to "Simulating Normalizing Constants: From Importance Sampling to Bridge
> Sampling to Path Sampling" by Gelman and Meng (1998) when introducing Eq. 8.**
>
> 2. > Could you explore and discuss the potential use of multiple particles (perhaps inspired by particle filters) to further enhance the performance of RA-SMC?
>
> This is a great question! We did perform preliminary studies on using multiple particles, primarily as a means to enable adaptive tempering schedules, which require population statistics.
> The difficulty arises because the initial $\theta$ sample is generated as a joint sample with $y$, which makes generating additional samples from the same posterior $p(\theta \mid y)$ non-trivial.
> We explored using approximate Bayesian computation (ABC) or MCMC to generate further samples from the same posterior as the initial sample in the path,
> but our preliminary results suggested that the additional computational cost and complexity of doing this significantly diminished the efficiency gains of the single particle method.
>
> However, we agree this is an important point worth discussing. We have **added a note to the conclusion** acknowledging the multi-particle extension as a direction for future work, while noting that the single-particle approach seems to hit a practical sweet spot in terms of accuracy and efficiency for the problems studied.
>
> 3. >  Given that "Sequential Monte Carlo" is widely used in the context of filtering, would it be possible to reconsider the use of this term for the proposed method? Alternatively, can you provide a clearer distinction between your method and traditional filtering-based SMC techniques?
>
> We thank the reviewer for raising this point about terminology, as the nomenclature in this area can be ambiguous and we considered this carefully during our work.
> The core idea of sampling through a sequence of tempered distributions is variously referred to as 'Annealed Importance Sampling' [3], 'Sequential Tempered MCMC' [1], and is related to the 'stepping-stone' algorithm [4].
>
> We ultimately chose the 'Sequential Monte Carlo' framework because we follow the convention of [2], which treats tempering as a primary application of the general SMC methodology. We feel this best positions our work within the broader class of modern Monte Carlo samplers.
>
> However, we agree that this should be made more explicit. To this end we have **added the following sentence the second paragraph of Section 2:**
>
> "We note here that in this work, 'sequential' refers to the progression of particles through a sequence of tempered distributions, as distinct from the temporal sequence in state-space and filtering applications."
>
> 4. > It would strengthen the paper to include a comparison with other modern methods for EIG estimation in BOED.
>
> We thank the reviewer for this important point. To begin addressing this, we have already run a new experiment during the rebuttal period comparing our method to a Prior Contrastive Estimation (PCE) baseline on the spring-mass model (as suggested by Reviewer Z8Ri), which confirmed the need for robust samplers for these problems. These results as well as a lengthier discussion on this point can be found in the rebuttal to Reviewer LaTK. We agree that a more comprehensive benchmark is a critical direction for future research, and we will highlight this explicitly in the conclusion of the final paper.
>
>
> [1] Catanach, T. A. and Beck, J. L. Bayesian updating and uncertainty quantification using sequential tem-363
> pered MCMC with the rank-one modified Metropolis algorithm. arXiv preprint arXiv:1804.08738,364
> 2018.
>
> [2] Chopin, N., Papaspiliopoulos, O., et al. An introduction to sequential Monte Carlo, volume 4.366
> Springer, 2020.367
>
> [3] Neal, R. M. Annealed importance sampling. Statistics and Computing, 11(2):125–139, April423
> 2001. ISSN 1573-1375. doi: 10.1023/A:1008923215028
>
> [4] Xie, W., Lewis, P. O., Fan, Y., Kuo, L., and Chen, M.-H. Improving marginal likelihood estimation439
> for Bayesian phylogenetic model selection. Systematic biology, 60(2):150–160, 2011.

---

> > ### Comment · Reviewer_2iaK · 2025-08-06
> >
> > Thank authors for their thorough and thoughtful responses. I am satisfied with the clarifications provided and maintain my original score: Strong Accept.

---

### Official Review · Reviewer_d56d · 2025-07-03

**Clarity:** 4
**Significance:** 3
**Originality:** 4
**Rating:** 5
**Confidence:** 3

**Summary:**

The manuscript proposes reverse--annealed sequential Monte Carlo for estimating the expected information gain (EIG) that drives Bayesian optimal experimental design.  The standard tempering route—used by sequential Monte Carlo samplers (Chopin 2002; Del Moral et al. 2006) and by many BOED works—starts with the prior and anneals forward toward the posterior with many particles.  Contrary to these forward tempering schemes, RA--SMC begins with an exact posterior draw obtainable in simulation--based design, then anneals backwards towards the prior while accumulating the log--evidence by a thermodynamics inspired integral.  The authors prove the estimator is asymptotically unbiased and demonstrate, on 4--6 dimensional case studies, that RA--SMC can match the accuracy of forward SMC using two orders of magnitude fewer likelihood evaluations.

**Questions:**

As far as I can understand, RA--SMC relies on the availability of an exact posterior sample, limiting applicability to fully simulator--based design; real--data or sequential--design contexts without simulators are excluded?

 Can likelihood--free posterior draws relax the ``exact posterior'' requirement and open RA--SMC to real--data?

**Ethical Concerns:**

["NO or VERY MINOR ethics concerns only"]

**Final Justification:**

Thanks to the authors for patiently addressing questions. I am satisfied with the clarifications provided and maintain my original score: Accept.

**Limitations:**

Yes

**Quality:**

4

**Strengths And Weaknesses:**

Strengths

The concept is elegant and practical.  It turns the necessity of prior sampling into an advantage by bootstrapping from a posterior sample naturally available in simulator settings. The theoretical development is solid. Lemma~4.2 establishes consistency under standard ergodicity conditions, and the temperature--schedule corollary offers practical guidance. The empirical study is careful. Each benchmark reports Monte Carlo error bars and raw likelihood budgets, allowing a transparent cost/accuracy comparison. The variance--decomposition analysis strongly justifies the single--particle regime, and the discussion of diagnostic uncertainty is refreshingly candid.

Prior work treats the posterior normalising constant Z(1) as unknown and therefore anneals forward. In BOED the authors observe that a joint prior draw  yields a known posterior density  p(θ∣y), so the reverse normalising constant  Z(0)=1 is known instead. Turning this observation into a practical estimator is, to my knowledge, new.

Weaknesses

All experiments remain in low--to--moderate dimensions with smooth likelihoods. Scalability to high--dimensional, multimodal or PDE--constrained models is not demonstrated.

Baseline coverage is narrow: annealed importance sampling, bidirectional Monte Carlo bounds, variational MI estimators and neural mutual--information surrogates are omitted despite being cited.

Runtime is measured only in likelihood calls; wall--clock timings and parallel scalability are not evaluated.

---

> ### Author Rebuttal · Authors · 2025-07-30
>
> We sincerely thank the reviewer for their positive assessment and insightful comments, which we address below.
>
> 1. >All experiments remain in low–to–moderate dimensions with smooth likelihoods. Scalability to high–dimensional, multimodal or PDE–constrained models is not demonstrated.
>
>
> The reviewer is correct that we focused on low-to-moderate dimensional problems.
> This was a deliberate choice to first establish the methodology and test its efficiency on problems known to be challenging due to their multimodal posterior distributions, as demonstrated for the spring-mass system (see Figure 3).
> We agree that demonstrating scalability to high-dimensional settings is an important step for future research. We are optimistic about our method's potential in this area and have **added the following to the conclusion to highlight high-dimensional scalability as a direction for future work:**
>
> "Because RA-SMC initializes from a true posterior sample, it may be less susceptible to the particle degeneracy issues that hinder traditional multi-particle SMC methods in high dimensions. Future research could explore ... studying the method's scalability in high dimensions."
>
>
> 2. >Baseline coverage is narrow.
>
> We thank the reviewer for this point, which was also raised by others. To strengthen the paper in line with this feedback, we have **run a new experiment during the rebuttal period comparing RA-SMC against Prior Contrastive Estimation on the spring-mass model** as suggested by Reviewer Z8Ri. Our results show that PCE, like standard Nested Monte Carlo, exhibits severe downward bias and is not effective on these challenging problems:
>
> |Design|0.2|0.4|0.6|0.8|1.0|1.2|1.4|1.6|1.8|2.0|
> |------|---|---|---|---|---|---|---|---|---|---|
> |EIG - PCE   |10.205|10.600|10.487|10.469|10.531|10.443|10.343|10.166|10.223|10.121|
> |SE - PCE |0.070 |0.061 |0.063 |0.074 |0.026 |0.044 |0.068 |0.063 |0.048 |0.077|
> |EIG - Forward SMC | 16.065 | 18.455| 20.269| 19.339|19.034|19.648|18.439|18.884|17.349|16.755|
>
> We chose forward SMC as our primary baseline because it represents the current state-of-the-art for asymptotically-exact EIG estimation in BOED. Our core contribution is showing that RA-SMC achieves comparable accuracy to this gold-standard method while requiring orders of magnitude fewer likelihood evaluations.
>
> A comprehensive comparison across other EIG estimators based on, e.g., variational inference or neural networks represents an excellent direction for future work. We have **added these new results to the appendix.**
>
>
> 3. >Runtime is measured only in likelihood calls; wall–clock timings and parallel scalability are not evaluated.
>
> We agree that parallel scalability is a valid concern. RA-SMC's single-particle nature limits parallelization compared to multi-particle forward SMC, which makes parallel scalability analysis less relevant for this method. However, we agree it would be good to clarify this in the manuscript. We report likelihood evaluations rather than wall-clock time since likelihood evaluation typically dominates runtime by orders of magnitude in simulation-based design. This provides a hardware-agnostic measure that directly translates to computational cost in any simulation-based BOED application, regardless of the specific forward model or computing environment.
>
> 4. >Does RA-SMC's reliance on exact posterior samples limit it to simulator-based design, excluding real-data contexts? Can likelihood-free posterior draws relax this requirement?
>
> Currently the RA-SMC estimator is limited to simulator-based problems.
> The question of whether likelihood-free methods can relax this requirement is very interesting. In principle, yes. If we can obtain approximate posterior samples, via for example approximate Bayesian computation (ABC), these could potentially serve as starting points for RA-SMC.
>
> We did some preliminary study on the efficacy of obtaining ABC samples for a different context (extending the method to utilize adaptive tempering).  Our preliminary experiments suggested that the approximation quality
> of the initial posterior samples directly affected RA-SMC's accuracy,
> and the efficiency gains were much less dramatic when accounting for the cost of obtaining the approximate posterior sample. However, our evaluations are still limited, so we are interested in studying this question further and have **added it as a direction for future work**.

---

> > ### Comment · Reviewer_d56d · 2025-08-08
> > **Thank you for the clarifications**
> >
> > This was a strong submission, and remains strong. I am confident that the extra work will make for a better paper.

---

### Official Review · Reviewer_LaTk · 2025-07-03

**Clarity:** 3
**Significance:** 2
**Originality:** 3
**Rating:** 5
**Confidence:** 3

**Summary:**

This paper proposes an algorithm for the estimation of the expected information gain (EIG) for Bayesian optimal experimental design via sequential Monte Carlo (SMC). One of the main quantities of interest for Bayesian experimental design algorithms is the marginal likelihood $p(y | d) = \int p(y|\theta, d) p(\theta) d\theta$ under a given design $d$.  The paper derives a reverse annealing scheme which samples $(y, \theta_*) \sim p(y, \theta)$ from the joint distribution treating the initial $\theta_* \sim p(\theta|y)$ effectively as a posterior sample, and guides the sampling towards the prior by decreasing a temperature parameter, while keeping track of the samples likelihoods. An estimate of the model evidence is them computable via a thermodynamic integral using Simpson's rule. Synthetic experiments evaluate the method against another SMC baseline.

**Questions:**

* In Definition 4.1, shouldn't the samples be $\{y_k, \theta_k)\}_{k=1}^K$? Currently, $d_k$ is in the sample set.

**Ethical Concerns:**

["NO or VERY MINOR ethics concerns only"]

**Final Justification:**

I've raised my score from 3 to 5 after clarifications and discussions from the rebuttal and other reviews. I'd encourage the authors to include the new experimental results and at least mention them in the main paper, add further details on the derivation of Eq. 8, provide clear references for Simpson's rule, and clarify the use of MCMC within the proposed SMC method, as discussed.

**Limitations:**

Limitations are discussed in Sec. 5.4.

**Paper Formatting Concerns:**

"acceptance rate using the feedback controller of (Catanach, 2017)" -> in-text citation should not be within parentheses.

**Quality:**

3

**Strengths And Weaknesses:**

### Strengths
* The paper is mostly well written.
* The methodology is based on interesting insights which might be useful for the design of other SMC-based EIG estimation algorithms.
* The algorithm seems to be relatively simple to implement compared to existing alternatives.
* Theoretical guarantees are provided on the asymptotic unbiasedness of the algorithm.

### Weaknesses
* The experimental evaluation involves a single SMC baseline, missing comparisons against other EIG estimators (e.g., Foster et al., 2019). Hence, it's hard to assess the significance of the contribution with respect to other methods for BOED.
* It is not clearly explained how MCMC is applied, even though it's a fundamental part of the algorithm.
* Eq. 8 might need further detail on its derivation, as it is not immediately clear that the integral on the right-hand side yields the log-marginal likelihood.
* I missed a reference for background on Simpson's rule.

---

> ### Author Rebuttal · Authors · 2025-07-30
>
> We are very grateful to Reviewer LaTK for the detailed feedback and for their positive comments on the paper's writing and novelty.
> We appreciate the constructive suggestions, which we address below.
>
> 1. > The experimental evaluation involves a single SMC baseline, missing comparisons against other EIG estimators ... it’s hard to assess the significance of the contribution with respect to other methods for BOED.
>
> We agree that positioning RA-SMC against a broader set of estimators would strengthen the paper.
> Our research here is focused on improving the computational efficiency of asymptotically-exact Monte Carlo estimators.
> Within this class of methods, we found that simpler approaches like Nested Monte Carlo can struggle on problems with complex posteriors.
> To address the reviewer's point under the time constraints of the rebuttal period and to verify that these problems are indeed challenging for non-tempering methods, we have run a **new experiment during the rebuttal period using the Prior Contrastive Estimation estimator on the spring-mass model** as suggested by Reviewer Z8Ri:
>
> |Design|0.2|0.4|0.6|0.8|1.0|1.2|1.4|1.6|1.8|2.0|
> |------|---|---|---|---|---|---|---|---|---|---|
> |EIG - PCE   |10.205|10.600|10.487|10.469|10.531|10.443|10.343|10.166|10.223|10.121|
> |SE - PCE |0.070 |0.061 |0.063 |0.074 |0.026 |0.044 |0.068 |0.063 |0.048 |0.077|
> |EIG - Forward SMC | 16.065 | 18.455| 20.269| 19.339|19.034|19.648|18.439|18.884|17.349|16.755|
>
> As shown in the table, the PCE estimates are stable but exhibit a severe downward bias, likely due to the highly informative data in these problems, failing to capture the true EIG or find the optimal design. The failure of both simpler baselines, NMC (as shown in Appendix A.3) and now PCE, provides evidence that these are nontrivial problems that require more complex estimators.
>
> Our contribution is therefore an order-of-magnitude efficiency gain for a powerful tool within this specific class of estimators. With that in mind, comparing RA-SMC to a strong forward SMC is the most logical and direct benchmark for our central claim. However, we agree that other classes of estimators, such as those based on variational inference or neural networks, offer a different set of compelling tradeoffs and a comprehensive comparison across these estimators is an excellent direction for follow-up work. We have **added the new PCE results to the table in Appendix A.3,** and we have **clarified the scope of our contribution in Section 2 (Related Work)** by adding the following paragraph:
>
> "The methods described above present a diverse set of trade-offs between computational cost, scalability, and asymptotic guarantees. This work specifically focuses on improving the efficiency of tempering-based Monte Carlo estimators, like AIS and SMC, which provide asymptotically-exact estimates crucial for high-fidelity applications, and we show that the RA-SMC estimator makes this powerful but expensive class of methods more practical. To that end, we first review the standard SMC framework, which forms the basis of our proposed approach."
>
> 2. > It is not clearly explained how MCMC is applied...
>
> We apologize for the lack of clarity here.
> At each step in the reverse annealing schedule,
> an MCMC kernel is used to generate a new sample from the current
> tempered distribution using the sample from the previous (higher temperature)
> distribution as its starting point.
> This provides the mechanism whereby $E_{\theta \sim p_{t_i}}\log p(y \mid \theta)$ is calculated for each $i = 0, \dots, N$.
> **We will add a short discussion in Section 4.2 to explicitly describe this process.**
>
> 3. > Eq. 8 might need further detail on its derivation...
>
> We agree that a short derivation would improve clarity.
> The identity follows from the Fundamental Theorem of Calculus, which we include below for convenience.
>
> \begin{aligned}
>     \log p(y)
>     &= \log p(y) - \log (1)
>     \\\\
>     &= \log (z_1(y)) - \log (z_0(y))
>     \\\\
>     &= \int_0^1 \frac{d}{dt} \log(z_t(y)) \,dt
>     \\\\
>     &= \int_0^1 \frac{1}{z_t(y)} \frac{d}{dt} z_t(y) \,dt
>     \\\\
>     &= \int_0^1 \frac{1}{z_t(y)} \frac{d}{dt} \int_{\theta}p(y \mid \theta)^t p(\theta)\,d\theta \,dt
>     \\\\
>     &= \int_0^1 \frac{1}{z_t(y)} \int_{\theta} \frac{d}{dt} p(y \mid \theta)^t p(\theta)\,d\theta \,dt
>     \\\\
>     &= \int_0^1 \frac{1}{z_t(y)} \int_{\theta} \log p(y \mid \theta) p(y \mid \theta)^t   p(\theta)\,d\theta \,dt
>     \\\\
>     &= \int_0^1 \int_{\theta} \log p(y \mid \theta) \frac{p(y \mid \theta)^t   p(\theta)}{z_t(y)}\,d\theta \,dt.
> \end{aligned}
>
> **We have added this derivation to Appendix A.1 and have added a reference to "Simulating Normalizing Constants: From Importance Sampling to Bridge
> Sampling to Path Sampling" by Gelman and Meng (1998) when introducing Eq. 8.**
>
> 4. > I missed a reference for background on Simpson's rule.
>
> Yes, thank you for catching this omission. **We have added citation for numerical integration methods** on line 158 where we first mention Simpson's rule.
>
> 5. > In Definition 4.1, shouldn't the samples be $(y_k, \theta_k)_{k=1}^K$? Currently, $d_k$ is in the sample set.
>
> Yes, this is a typo. Thank you for pointing it out. We have corrected it in the manuscript.
>
> 6. > ``acceptance rate using the feedback controller of (Catanach, 2017)'' -> in-text citation should not be within parentheses.
>
> We have corrected this in the manuscript.

---

> > ### Comment · Reviewer_LaTk · 2025-08-04
> >
> > I'd like to thank the authors for addressing my main concerns. Given the clarifications, additional results and other (positive) reviews and discussions, I'll raise my score. I'd strongly encourage the authors to revise the manuscript based on the feedback from these discussions, which should significantly strengthen the paper.

---

### Decision · Program_Chairs · 2025-09-17

**Decision:**

Accept (poster)

**Comment:**

This paper presents an interesting and novel approach to efficiently estimating EIG (within Bayesian experimental design) using Annealed Sequential Monte Carlo. All reviewers agreed on the significance and novelty of the work. I suggest that the authors incorporate the additional experiments and clarifications raised during the discussions with reviewers in the rebuttal into the final version of the paper.